# Stereoselective synthesis of sulfur-containing β-enaminonitrile derivatives through electrochemical Csp³–H bond oxidative functionalization of acetonitrile

Tian-Jun He[1], Zongren Ye[2], Zhuofeng Ke [2] & Jing-Mei Huang [1]

Incorporation of nitrile groups into fine chemicals is of particular interest through C(sp³)–H bonds activation of alkyl nitriles in the synthetic chemistry due to the highly efficient atom economy. However, the direct α-functionalization of alkyl nitriles is usually limited to its enolate chemistry. Here we report an electro-oxidative C(sp³)–H bond functionalization of acetonitrile with aromatic/aliphatic mercaptans for the synthesis of sulfur-containing β-enaminonitrile derivatives. These tetrasubstituted olefin products are stereoselectively synthesized and the stereoselectivity is enhanced in the presence of a phosphine oxide catalyst. With iodide as a redox catalyst, activation of C(sp³)–H bond to produce cyano-methyl radicals proceeds smoothly at a decreased anodic potential, and thus highly chemoselective formation of C–S bonds and enamines is achieved. Importantly, the process is carried out at ambient temperature and can be easily scaled up.

[1] Key Laboratory of Functional Molecular Engineering of Guangdong Province, School of Chemistry and Chemical Engineering, South China University of Technology, 510640 Guangzhou, China. [2] School of Materials Science and Engineering, PCFM Lab, Sun Yat-sen University, 510275 Guangzhou, China. These authors contributed equally: Tian-Jun He, Zongren Ye.  Correspondence and requests for materials should be addressed to J.-M.H. (email: chehjm@scut.edu.cn)

Nitriles are widely found in pharmaceuticals, natural products, and materials[1–4]. Introduction of nitrile groups onto target frameworks by C–H functionalization of C(sp$^3$)–H bonds of simple aliphatic nitriles is of particular interest in the synthetic chemistry. Early research has focused on the C–H activation of alkyl nitriles using stoichiometric amounts of transition metal salts (such as Ru, Rh, Ni, Fe, etc.)[5–9]. On the other hand, the direct α-functionalization of alkyl nitriles is usually limited to its enolate chemistry, which requires a strong base for its formation[10–15]. Recently, free radical-initiated α-C–H functionalization of alkyl nitriles has attracted attention (Fig. 1a)[16,17]. Nevertheless, these methods require excess equivalents of strong oxidants (peroxides), metal-based oxidants (Ag$^+$, Mn$^{3+}$), or single-electron transfer reagents (diazonium salts). In addition, all of these works were carried out at elevated temperatures. Therefore, methods for the mild and environmentally friendly activation of alkyl nitriles are still highly desirable.

Electrochemical anodic oxidation represents an attractive and environment-friendly synthetic strategy to solve lingering problems in organic chemistry[18–33]. Particularly, the indirect electrolysis, in which a redox catalyst is utilized as the electron shuttle, achieves higher energy efficiency and different selectivity[34–43]. In continuation of our interest in the development of electrochemical methods to organic synthesis[44–47], herein, we report the electrochemical C(sp$^3$)–H bond oxidative functionalization of acetonitrile mediated by potassium iodide to synthesize sulfur-containing β-enaminonitrile derivatives highly efficiently in one pot (Fig. 1b).

## Results

**Reaction optimization.** To initiate the investigation, *p*-fluorothiophenol (**1aa**) and acetonitrile were chosen as the substrates to test the reaction. Under a galvanostatic condition at 10 mA in an undivided cell, the reaction of **1aa** and acetonitrile with 10 mol% citric acid, 20 mol% 1,2-bis(diphenylphosphino)ethane (DPPE), and 50 mol% KI gave a 96% yield of the desired product as a pair of isomers of **2aa** ($Z/E = 19:1$, Table 1, entry 1) and a simple column chromatography separation could give the pure *Z*-isomer. Both the yield and stereoselectivity were reduced when the reaction was carried out in the absence of citric acid (Table 1, entry 2). The stereoselectivity decreased to 12:1 without DPPE (Table 1, entry 3). When the reaction was performed in the absence of KI, no desired product was obtained, which indicated

that KI played a key role in this selective electro-oxidative reaction (Table 1, entry 4). Replacing citric acid with acetic acid led to a lower yield at 79% (Table 1, entry 5). No desired product was detected when *t*-BuOK was added instead of citric acid (Table 1, entry 6). When the temperature decreased to 10 °C, a 67% yield of **2aa** was found (Table 1, entry 7). Heating the reaction mixture to 50 °C or 70 °C resulted in poor stereoselectivities (Table 1, entries 8 and 9). Inferior reaction yield and stereoselectivity were obtained when *N,N*-dimethylformamide (DMF) was used as a co-solvent (Table 1, entry 10). The influence of the electrodes was also studied. Replacing either the Pt minigrid anode or the Pt wire cathode by a Pt foil led to a poor result (Table 1, entries 11 and 12). The increase of the electric current caused a lower yield and stereoselectivity (Table 1, entry 13). No desired product could be detected when the reaction was carried out at a current lower than 5 mA (Table 1, entry 14). Screening of acids or bases, redox catalysts, ligands, electrolytes, and electrode materials were also studied (See Supplementary Tables 1–5).

**Substrate scope.** With the optimized conditions defined (Table 1, entry 1), the scope of phenylthiols/thiols was probed. As shown in Fig. 2, the reactions of various phenylthiols/thiols proceeded smoothly and the desired products were obtained in good to excellent yields with good stereoselectivities in most cases. First, the reactivity of phenylthiols with substituents on the benzene ring was studied. In general, both electron-donating and electron-withdrawing groups with different substitution patterns (*para*-, *meta*-, and *ortho*- substitutions; mono and multi substitutions) were tolerated in this reaction. Aryl thiols bearing fluoro, chloro, bromo, methyl, and methoxy groups could give the desired products in excellent yields (90–96%) and good selectivities. In addition, other fluorine-containing phenylthiols, such as trifluoromethyl phenylthiol and pentafluoro phenylthiol, were compatible with this protocol, affording **2ad** and **2f** in 78 and 79% yields, respectively. Somewhat steric effects were observed with the functional groups on the *ortho*-position (**2bb–2be**). Interestingly, oxidatively labile functional groups, such as amino and hydroxy, were tolerated in this transformation to produce the corresponding products (**2ae** and **2c**) in 24 and 47% yields, respectively. The present method could also be applied to diphenyl disulfide (**2g**).

Naphthyl and heteroaromatic thiols (**2 h**, **2i–2k**) were effective for this reaction. Notably, the substrate scope could also

Strategies for the formation of nitrile-containing alkyl radical:

**a** Previous work: traditional organic chemistry or photochemistry

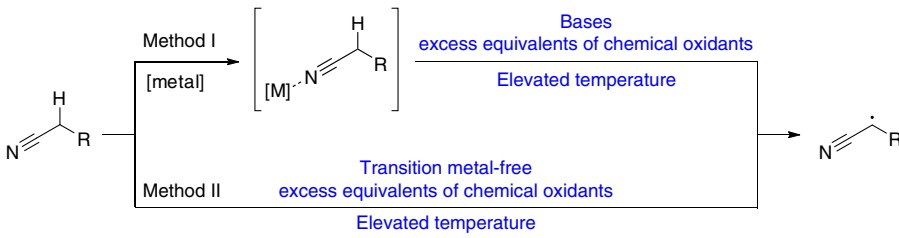

**b** This work: electrochemistry

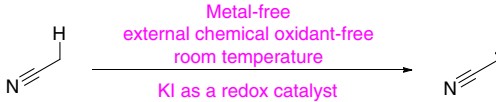

**Fig. 1** Strategies for the formation of nitrile-containing alkyl radicals. **a** The formation of nitrile-containing alkyl radicals under traditional organic chemical or photochemical conditions. **b** The formation of cyanomethyl radicals under electrochemical conditions

**Table 1 Optimization of reaction conditions[a]**

| Entry | Variation from the standard conditions | Yield (%)[b] | Z/E[b] |
|---|---|---|---|
| 1 | None | 96 | 19:1 |
| 2 | Without citric acid | 44 | 10:1 |
| 3 | Without DPPE | 95 | 12:1 |
| 4 | Without KI | 0 | |
| 5 | Acetic acid instead of citric acid | 79 | 13:1 |
| 6 | t-BuOK instead of citric acid | 0 | |
| 7 | 10 °C | 67 | 10:1 |
| 8 | 50 °C | 99 | 12:1 |
| 9 | 70 °C | 99 | 12:1 |
| 10 | MeCN/DMF = 1:1 | 80 | 9:1 |
| 11[c] | Pt foil as an anode | 71 | 9:1 |
| 12[c] | Pt foil as a cathode | 66 | 9:1 |
| 13 | 15 mA, 3 h | 84 | 13:1 |
| 14 | 5 mA, 8 h | 0 | |

DPPE 1,2-bis(diphenylphosphino)ethane, DMF N,N-dimethylformamide, [19]F NMR fluorine-19 nuclear magnetic resonance
[a]Standard conditions: **1aa** (0.5 mmol), citric acid (10 mol%), DPPE (20 mol%), KI (50 mol%), MeCN (5 mL), with 0.1 M n-Bu$_4$NClO$_4$ as electrolyte. A Pt minigrid electrode (52 mesh, 1 × 1.5 cm$^2$) as an anode and a Pt wire (diameter = 0.5 mm, height = 2.0 cm) as a cathode, an undivided cell, constant current = 10 mA, 4 h, room temperature, 3.0 F mol$^{-1}$
[b]Yields and Z/E ratios were determined by [19]F NMR analysis of the crude reaction mixture using fluorobenzene as the internal standard
[c] Pt foils (1.0 × 1.5 cm$^2$)

be extended to alkyl mercaptans. The primary thiols, 2-phenylethanethiol and 1-hexanethiol, could be transformed to the corresponding products (**2l** and **2m**) in 53 and 24% yields, respectively. The secondary thiol, that is, cyclohexanethiol gave the product (**2n**) in a yield of 5% only. Unfortunately, no **2o** was detected when benzyl mercaptan was used as the reactant. Finally, it was delightful to find that diphenyl diselenide (**2p**) and dimethyldiselenide (**2q**) could be converted effectively to the corresponding products in excellent yields and good selectivitities.

The absolute stereochemistry of one of the products (**Z**)-**2ba** was determined by X-ray crystallographic analysis (Fig. 2). The stereochemistry of other products was determined on the basis of the similarities of the polarities and the [1]H NMR and [13]C NMR chemical shifts.

Next, the synthetic utility of this methodology was further investigated. First, the gram-scale synthesis of **2aa**, **2ah**, and **2bb** was performed and the desired products were obtained in the yields of 78%, 81%, and 65%, respectively. Second, 4H-1,4-benzothiazine scaffolds were obtained by copper-catalyzed cyclization of **2bb** and the corresponding derivatives (**3a** and **3b**) in the conversion yields of 70–87%. Notably, 4H-1,4-benzothiazine scaffolds are widely used in pharmaceutical chemistry due to their activities of antimicrobial, anticancer, and so on (Fig. 3).

## Discussion

Further studies were carried out to gain more insights into the reaction mechanism. First, when a radical scavenger, TEMPO (tetramethylpiperidine N-oxyl) or BHT (butylated hydroxytoluene), was added into the reaction mixture under the standard conditions, only a trace of desired product **2aa** was detected (Fig. 4a), and thus a radical nature of the transformation was implied. On the other hand, 3-aminocrotononitrile **4** was detected by gas chromatography–mass spectrometry (GC–MS) analysis

during the above two reaction processes. Subsequent investigation demonstrated that the reaction started from **4** could produce the desired product **2aa** (Fig. 4b), while no **2ab** was obtained from (4-chlorophenylthio)acetonitrile **5** (Fig. 4c). Therefore, it was confirmed that the first step of this tandem reaction is an acetonitrile self-condensation to produce **4**.

Next, the formation of **4** was investigated. The pH value was monitored and it showed that the pH value increased steadily from 2 to 6 (see Supplementary Figure 1). Obviously, it was not a traditional Thorpe-type self-condensation through the $^-$CH$_2$CN, which usually occurs under strongly basic conditions[48–50]. On the other hand, a radical trapping adduct, **6**, was detected (GC–MS analysis) by the use of 1,1-diphenylethene as the radical inhibitor, which suggested the intermediacy of cyanomethyl radical (Fig. 5a). Notably, the reaction did not proceed to afford the desired product **2aa** when acetonitrile was replaced by iodoacetonitrile (Fig. 5b). Moreover, using iodoacetonitrile instead of KI could not generate the desired product **2aa** even if acetonitrile was used as a solvent (Fig. 5b). These results rule out the formation of the intermediate ICH$_2$CN in the early stage of the reaction.

No desired reaction occurred in the absence of KI under the standard conditions (Table 1, entry 4). Further studies showed that in the absence of thiol, **4** could be obtained (Fig. 6a), while no **4** was detected without the addition of KI into the above reaction mixture. It demonstrated that KI played a crucial role in the formation of **4**. It was observed that the production of **4** needed the galvanic current, but the standard reaction could proceed in the dark (Fig. 6b). Hence, the formation of **4** might have been catalyzed by an iodine species that was generated by anodic oxidation, instead of photoexcitation, from KI[51,52]. Next, to explore the actual iodine species in the reaction, several different stoichiometric amounts of iodine sources were employed in the model reaction without the current. No **4** could be detected when

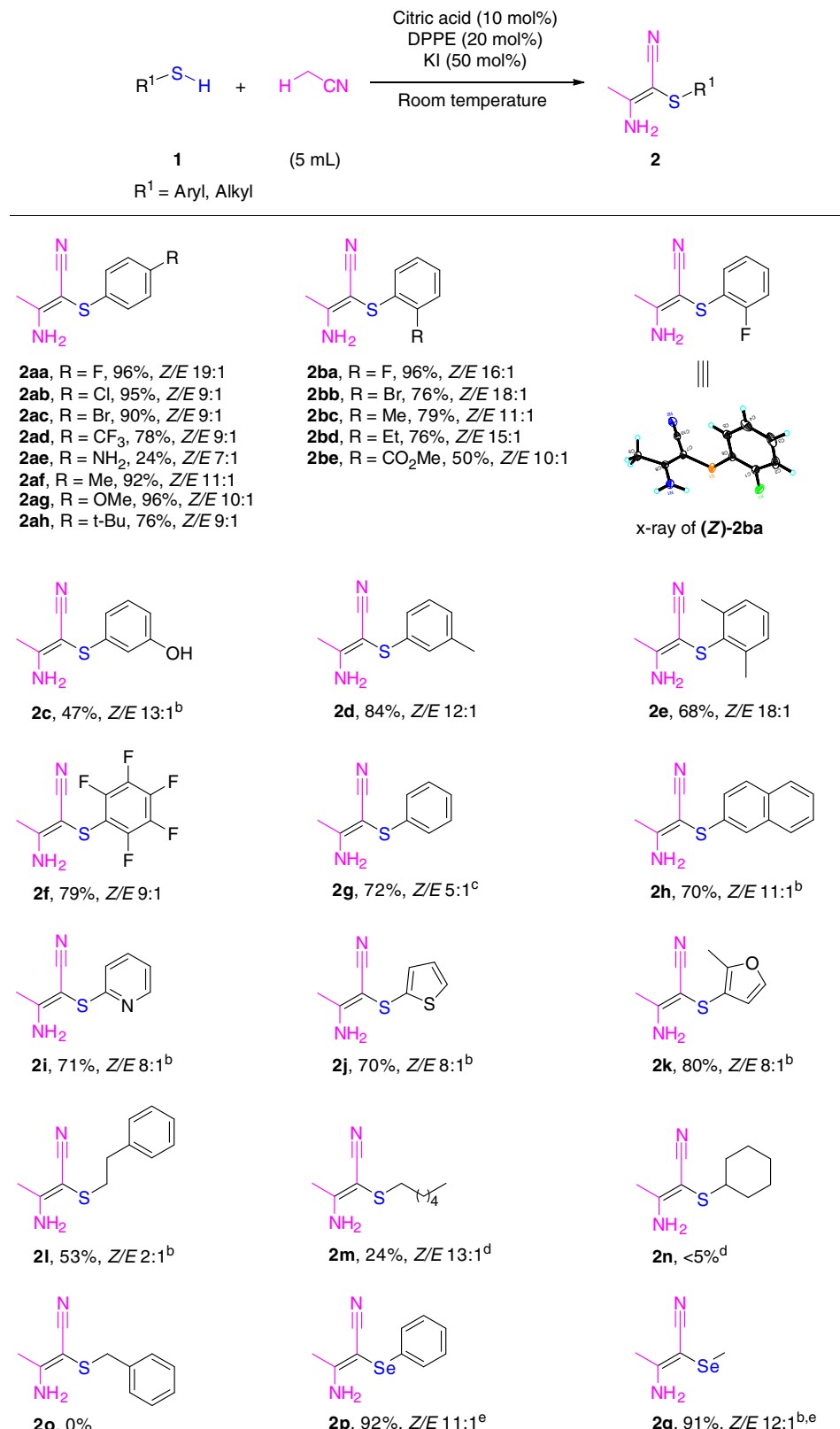

**Fig. 2** Substrate scope of thiols. [a]Standard conditions: **1** (0.5 mmol), citric acid (10 mol%), 1,2-bis(diphenylphosphino)ethane (DPPE) (20 mol%), KI (50 mol%), MeCN (5 mL), with 0.1 M n-Bu$_4$NClO$_4$ as electrolyte. A Pt minigrid electrode as an anode and a Pt wire as a cathode, an undivided cell, constant current = 10 mA, 4 h, room temperature. Isolated yields are shown. Z/E ratios were determined by fluorine-19 nuclear magnetic resonance ($^{19}$F NMR) or proton nuclear magnetic resonance ($^1$H NMR analysis). [b]KI (60 mol%), reaction time: 6 h. [c]Phenyl disulfide as a substrate. [d]KI (60 mol%), reaction time: 10 h. [e]Diselenides as substrates

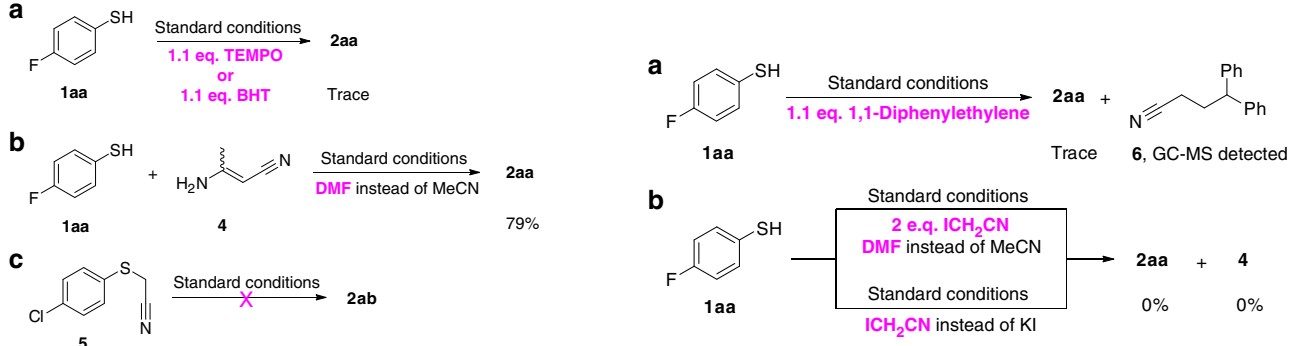

**Fig. 3** Gram-scale synthesis and product transformations. Reaction conditions: (i) acetyl chloride, Et₃N, CH₂Cl₂, 0 °C to reflux, 12 h, 85%; BnBr, NaH, dry N,N-dimethylformamide (DMF), N₂, 0 °C to r.t., 4 h, 79%. (ii) CuI, K₂CO₃, trans-N,N′-dimethylcyclohexane-1,2-diamine, N,N′-dimethylethylenediamine, toluene, N₂, 120 °C, 15 h, conditions to be optimized. [a]Conversions: **3c**, 60%; **3d**, 72%; **3e**, 62%

**Fig. 4** Mechanistic studies on the reaction. **a** Reactions by adding radical scavengers (tetramethylpiperidine N-oxyl (TEMPO) or butylated hydroxytoluene (BHT)). **b** The reaction between **1aa** and **4**. **c** The reaction between **5** and acetonitrile

**Fig. 5** Studies on the pathway for the formation of intermediate **4**. **a** Radical trapping experiment by 1,1-diphenylethylene. **b** The reaction between **1aa** and iodoacetonitrile

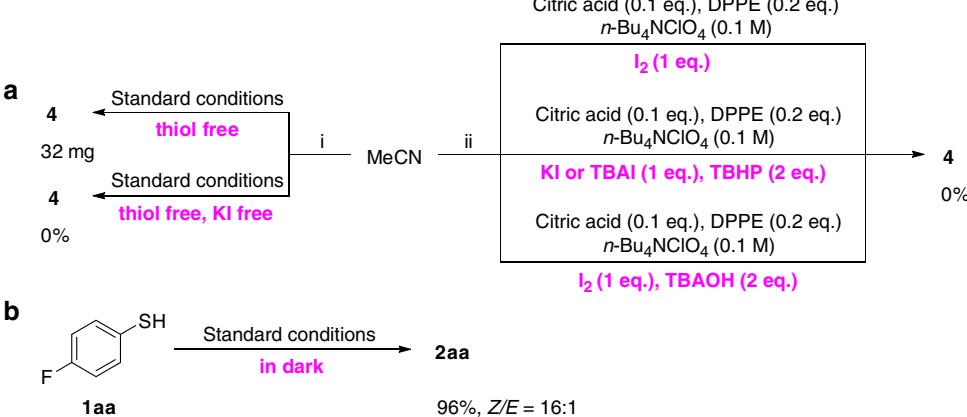

**Fig. 6** Studies on the active catalytic species for the formation of intermediate **4**. **a** I. Studies on the formation of **4**; II. the exploration of the iodine species. **b** The standard reaction run in dark

$I_2$ was applied (Fig. 6a). It has been reported that quaternary ammonium hypoiodite $[n\text{-}Bu_4N]^+[IO]^-$ or iodite $[n\text{-}Bu_4N]^+[IO_2]^-$ could abstract a hydrogen atom from a $C(sp^3)-H$ to

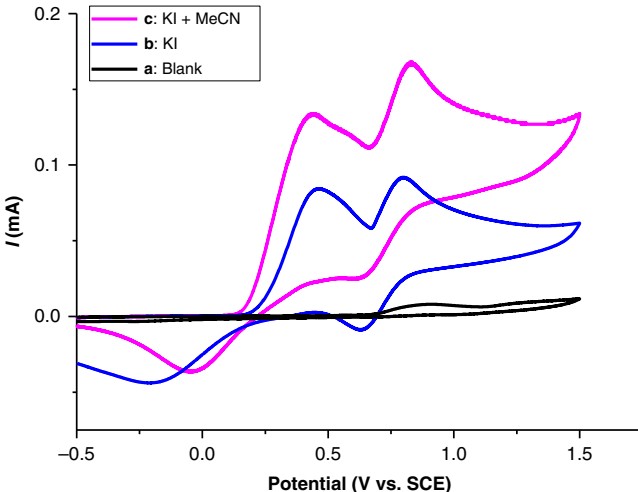

**Fig. 7** Cyclic voltammograms of 0.1 M $n\text{-}Bu_4NClO_4$ solution in *N,N*-dimethylformamide (DMF) at room temperature. **a** None; **b** KI (50 mmol L$^{-1}$); **c** KI (50 mmol L$^{-1}$) + MeCN (1 mL). The voltammogram was obtained with Pt wire as an auxiliary electrode and a saturated calomel electrode (SCE) as a reference electrode. The scan rate was 0.1 V s$^{-1}$ on a platinum disk electrode ($d = 2$ mm)

produce a radical[53]. However, the subsequent investigations using KI/TBHP, TBAI/TBHP, or $I_2$/TBAOH system, which have been reported to generate hypoiodite or iodite, gave no **4** (Fig. 6a). Thus, we hypothesized that the active iodine radical[54–58], which was in situ generated from the anodic oxidation, was able to abstract one hydrogen atom from acetonitrile to form the cyanomethyl radical.

Cyclic voltammetry studies (see Supplementary Figure 2) showed that KI (Fig. 7, curve b) exhibited two pairs of typical redox waves, with the oxidation peaks at 0.46 V (Ox$_1$) and 0.80 V (Ox$_2$) vs. SCE. After acetonitrile was introduced, obvious catalytic currents were detected; the peak currents of Ox$_1$ and Ox$_2$ dramatically increased from 84 to 134 and 92 to 168 μA, respectively (Fig. 7, curve c). Therefore, it was suggested that KI was employed as a redox catalyst in this indirect electrolysis process.

On the basis of these above results, a plausible mechanism was proposed (Fig. 8). The reaction sequence began with the in situ generation of an iodine radical on the anode and the iodine radical abstracted one hydrogen atom from acetonitrile to form the cyanomethyl radical **7**. Addition of **7** to another molecule of acetonitrile furnished intermediate **8**[59]. The α-imine radical intermediate **10** was obtained by a 1,3-hydrogen transfer[60–63] of iminyl radical **8**. Meanwhile, thiol **1aa** could be oxidized by the redox catalyst or by the anode directly to afford a sulfur radical **12**, which underwent dimerization to generate a disulfide **13**[64–66]. Thus, radical intermediate **10** could substitute with the disulfide **13** or couple with the sulfur radical **12** directly to produce imine **11**, which could tautomerize to give the desired product **2aa** in the presence of the acid catalyst. However, another pathway cannot be ruled out. Tautomerization of **10** to the corresponding

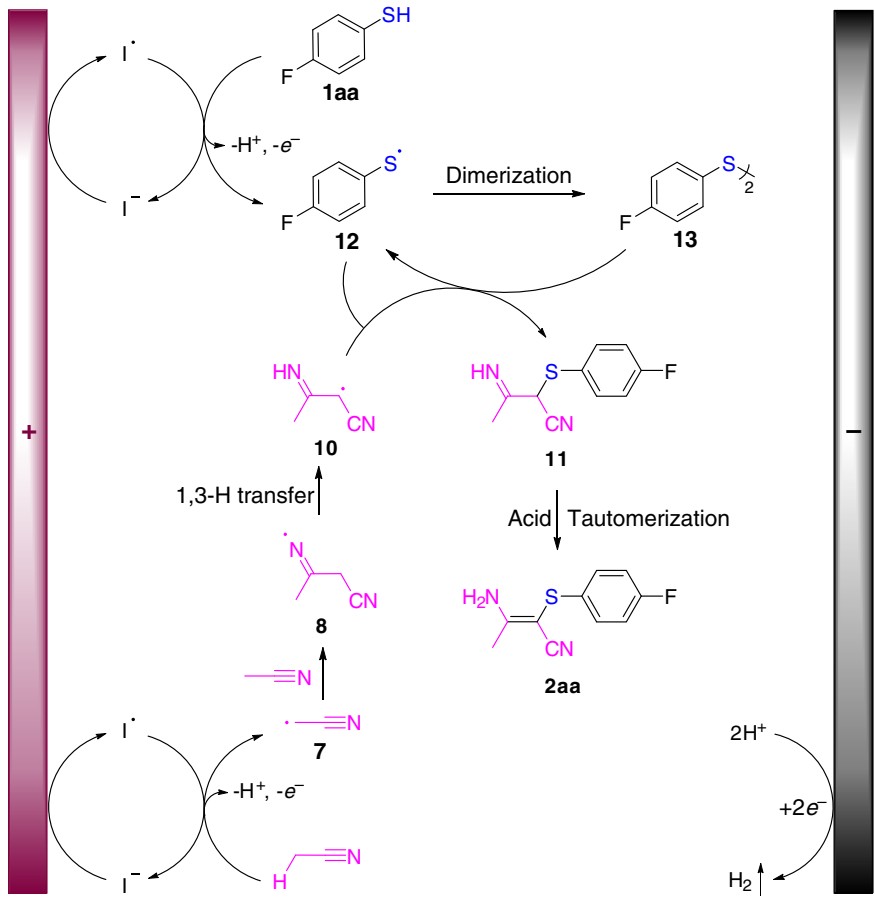

**Fig. 8** Proposed mechanism. Proposed reaction mechanism involves indirect anode oxidation of acetonitrile to cyanomethyl radical **7**, addition to acetonitrile, 1,3-H transfer to produce **10**, reaction of **10** with **12** or **13**, and tautomerization to furnish the final product **2aa**

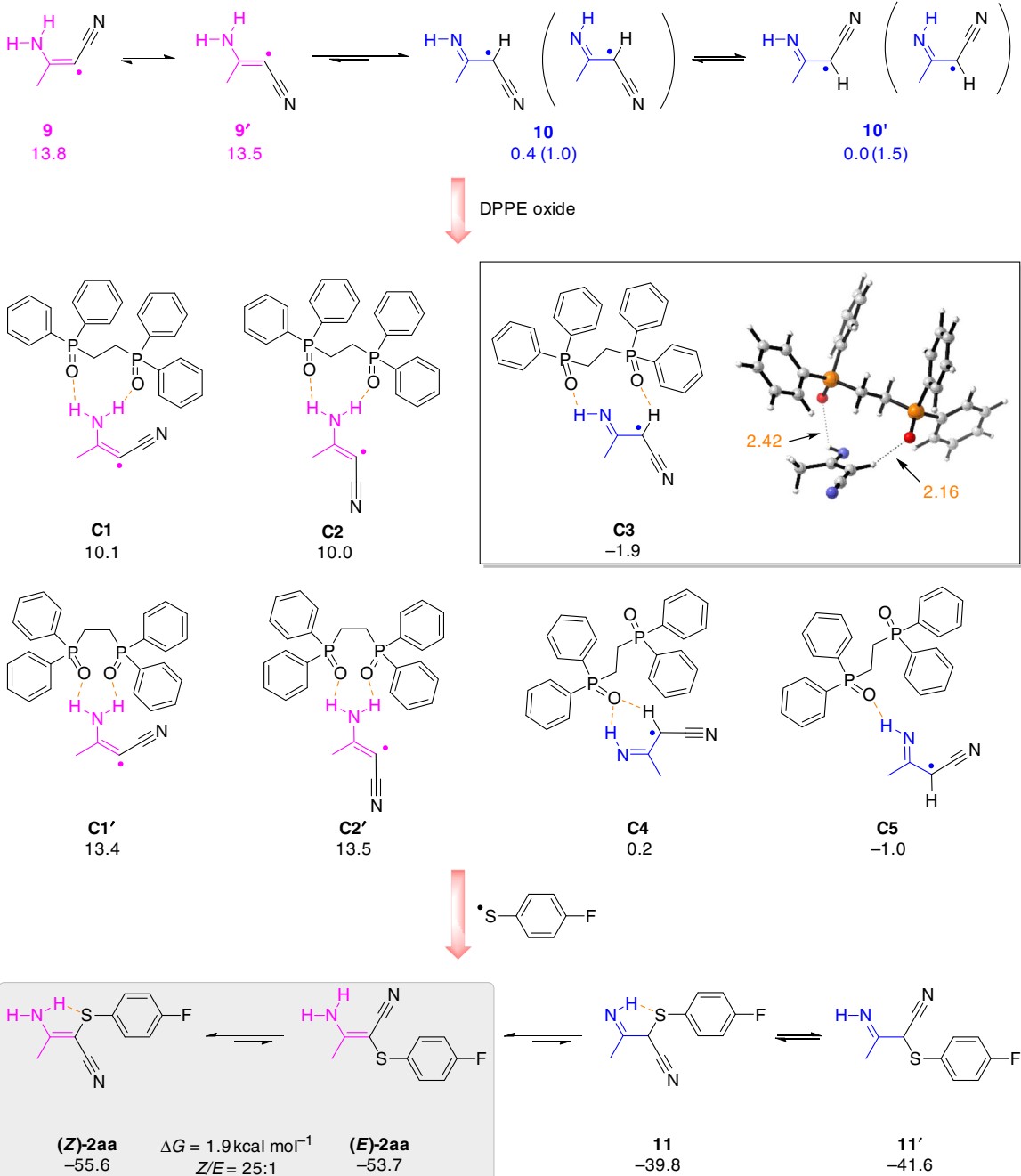

**Fig. 9** Density functional theory (DFT) study of the key intermediates and the calculated structures (Å, red) for complex **C3**. DFT study shows imine radical **10** can be coordinated with 1,2-bis(diphenylphosphino)ethane (DPPE) oxide to form complex **C3** via hydrogen bonding. The result suggests the important role of DPPE oxide in stabilizing the imine radical **10**, facilitating the formation of *Z* product and its tautomerization to final product **(Z)-2aa**

enamine radical **9** was followed by the substitution with the disulfide **13** or the coupling with the sulfur radical **12** directly to form the desired product **2aa**. Concomitantly, cathodic reduction of protons led to the release of $H_2$.

It was observed that the selectivity increased when DPPE was added. Studies showed that DPPE was in situ oxidized to 1,2-ethylene bis(diphenylphosphine oxide) on the anode (see Supplementary Discussion). Pre-oxidation of DPPE on the anode before the main reaction occurred the same yield and selectivity. We further carried out a density functional theory (DFT) calculations to provide insights into the mechanism (Fig. 9). DFT results indicate that imine type radical intermediates (**10** or **10′**) are more stable than the enamine type radicals (**9** or **9′**). The

imine radical would interact with DPPE oxide to form complexes **C3–C5**, among which **C3** is the most stable one with the calculated formation energies of −1.9 kcal mol$^{-1}$. The complex **C3** can stabilize the yielded radical and facilitate the C–S bond formation in *Z* configuration. It should be noted that the enamine type products (**2aa**) are more stable than imine types (**11** and **11′**). Therefore, the formed imine products would tautomerize to give the desired product **2aa**, in which the **(Z)-2aa** is more stable than the **(E)-2aa** by 1.9 kcal mol$^{-1}$. The predicted *Z:E* is around 25:1, which is in excellent agreement with our experimental observation (19:1). Considering the relative stability between **(Z)-2aa** and **(E)-2aa**, the reaction should be thermodynamic control. DFT results suggest the important role of DPPE oxide in stabilizing the

imine radical **10**, facilitating the formation of *Z* product and its tautomerization to final product (*Z*)-**2aa**.

In conclusion, we have developed a radical-initiated C(sp³)–H bond oxidative functionalization of acetonitrile through a KI-mediated indirect anodic oxidation. A wide range of aromatic/aliphatic mercaptans bearing various functional groups could participate in the reactions with acetonitrile to afford sulfur-containing β-enaminonitrile derivatives with concomitant generation of (*Z*)-tetrasustituted olefins. The high chemoselectivities and good stereoselectivities of the reactions could be achieved under metal-free, external chemical oxidant-free conditions. Further investigations into the mechanistic details and synthetic applications are currently underway in our laboratory.

## Methods

**Representative procedure for the synthesis of 2aa**. Into a round bottom flask was added thiol **1aa** (0.5 mmol, 1.0 equiv), KI (50 mol%), citric acid (10 mol%), and DPPE (20 mol%). MeCN (5 mL) with $n$-Bu$_4$NClO$_4$ (0.1 M) as an electrolyte was then added. The resulting solution was electrolyzed with a Pt minigrid electrode (52 mesh, $1 \times 1.5$ cm²) as anode and a Pt wire (diameter = 0.5 mm, height = 2.0 cm) as cathode, under a constant current (10 mA) in an undivided cell at room temperature for 4 h. After electrolysis, the mixture was quenched by water and extracted with ethyl acetate ($3 \times 10$ mL). The combined organic layer was washed with brine (10 mL) and dried over Na$_2$SO$_4$. The ratio of (*Z*)-**2aa** and (*E*)-**2aa** was determined by ¹⁹F NMR (*Z/E* ratio = 19:1) of the crude mixture. ¹⁹F NMR (377 MHz, CDCl$_3$) δ −115.97 (major), −116.73 (minor). The mixture of (*Z*)-**2aa** and (*E*)-**2aa** was obtained by a column chromatography separation of the crude mixture on silica gel (petroleum ether/ethyl acetate = 2:1), colorless oil, 100.0 mg, 96%. And, a further column chromatography separation could give the pure *Z*-isomer.

**Procedure for the scale-up synthesis of 2aa**. Into a round bottom flask was added KI (50 mol%), citric acid (10 mol%), and DPPE (20 mol%). MeCN (60 mL) with $n$-Bu$_4$NClO$_4$ (0.1 M) as an electrolyte was added. Thiol **1aa** (6 mmol, 1.0 equiv) was then introduced. The resulting solution was electrolyzed with a Pt minigrid electrode (52 mesh, $1 \times 1.5$ cm²) as anode and a Pt wire (diameter = 0.5 mm, height = 2.0 cm) as cathode, under a constant current (10 mA) in an undivided cell at room temperature. After 50 h, the mixture was quenched by water and extracted with ethyl acetate ($3 \times 30$ mL). The combined organic layer was washed with brine (20 mL) and dried over Na$_2$SO$_4$, filtered, and concentrated in vacuo. The mixture of (*Z*)-**2aa** and (*E*)-**2aa** was obtained by a column chromatography separation of the crude mixture on silica gel (petroleum ether/ethyl acetate = 2:1), colorless oil, 0.97 g, 78%, *Z/E* ratio = 19:1.

## Data availability

The X-ray crystallographic coordinates for structures reported in this article have been deposited at the Cambridge Crystallographic Data Center (CCDC), under deposition number CCDC 1849256 ((*Z*)-**2ba**). The data can be obtained free of charge from The Cambridge Crystallographic Data Center via http://www.ccdc.cam.ac.uk/data_request/cif. For full characterization data including NMR spectra of new compounds and experimental details, see the Supplemental Information. Any further relevant data are available from the authors upon reasonable request.

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

## Acknowledgements

We are grateful to the National Natural Science Foundation of China (Grant Nos. 21672074 and 21372089) for financial support.

## Author contributions

J.-M.H. directed the project. J.-M.H. and T.-J.H. designed the project. T.-J.H. performed the experiments and analyzed the data. Z.K. and Z.Y. performed DFT calculations and analyzed the data. All authors contributed to scientific discussion and co-wrote the manuscript.

## Additional information

**Competing interests:** The authors declare no competing interests.

