## [Peer Review File · Nature Communications]

Reviewers' comments:

Reviewer #1 (Remarks to author):

Comments:

During the past years, the field of organic electrocatalysis has undergone a renaissance and has attracted considerable attention from the chemistry community for the development of sustainable bond-forming platforms. The manuscript by He and Huang reports a method for electro-oxidative C(sp³)-H bond functionalization of alkyl nitriles to furnish sulfur-containing β -enaminonitrile derivatives. I think that the manuscript describes a rather unique chemical transformation that distinguishes the work from a majority of previously developed electrocatalytic bond-forming protocols. Furthermore, the developed transformation provides potentially synthetically valuable products; however, the synthetic utility should be further highlighted. Therefore, I believe that the manuscript could be a nice addition to researchers working in the electrocatalysis community, and therefore support publication of the work in Nature Communications if the concerns and technical issues described below can be successfully addressed.

Concerns:

- 1) Can the developed protocol be extended to other alkyl nitriles or is it limited to acetonitrile?
- 2) The synthetic utility of the products should be further highlighted, i.e. what can, for example, compound 2aa be transformed into?
- 3) The role of 1,2-bis(diphenylphosphino)ethane (DPPE) needs to be further investigated/clarified in order to rule out coordination to free metal ions, i.e. acting as a ligand.
- 4) Also, the authors state that "it was tentatively proposed that the corresponding oxide of DPPE might have coordinated with the intermediate through the hydrogen-bonding interactions, which renders the formation of Z-isomer more favorable". However, it is not clear how/in what way the potentially oxidized DPPE facilitates formation of the Z-isomer. Please highlight this with a figure and if possible provide, for example, calculations that support this hypothesis.

Comments on style/writing issues:

- 5) I strongly suggest the authors to use colors in the figures as this would make the figures more attractive.
- 6) Some language corrections are necessary (please see the attached pdf-file).

Reviewer #2 (Remarks to the Author):

This manuscript reports a radical initiated C(sp³)-H bond oxidative functionalization of alkyl nitriles through a KI mediated indirect anodic oxidation. High chemoselectivities and good stereoselectivities of the reactions could be achieved under the metal-free, external chemical oxidant-free conditions. A reliable mechanism is proposed after some control experiments and cyclic voltammetry experiments were conducted. The results described by the authors are informative and

interesting to Nature Communications readers, particularly those working on the electroorganic chemistry. Based on the content and quality of this paper, I think this work is suitable for Nature Communications however some issues should be addressed before the accepting this manuscript for publication.

1) The reaction conditions have been optimized (constant current conditions - it would be useful to also report the potential range from the beginning to the end of these electrosyntheses) in an undivided cell. Some information on the electrode choice is required. Can the reaction also be performed with a carbon working electrode? More cheap electrodes should be examined.

2) In the proposed mechanism, radical intermediate 9 could substitute with the disulfide 11 to produce an imine 12, the authors should give the corresponding evidence to prove this process. A reaction of disulfide 11 with acetonitrile should be conducted.

3) In order to make the mechanism more convincing, authors should carry out more cyclic voltammetry (CV) experiments. For example, the CV experiments of a mixture of 1aa, acetonitrile, and KI should be done.

4) Table 1, entry 2 and entry 3, the author's description does not conform to the table.

Reviewer #3 (Remarks to the Author):

In the present manuscript, Huang and He develop a method for electro-oxidative C(sp³)-H bond functionalization of acetonitrile to synthesize sulfur-containing beta-enaminonitriles. While the work is interesting, it is not breaking new ground at a level for publication in Nature Communications. Although several functional groups were contained in the products, such products are lack of universality. As it stands the work is incomplete in relation to limited utility of the processing of aliphatic thiols. Except for acetonitrile, no other alkyl nitriles were tested in the manuscript. As a research on electrochemical synthesis, the authors do not explain why the reaction cannot occur when LiClO₄ and NaClO₄ were used as the electrolyte. This is not a common phenomenon. Most importantly, the manuscript contains insufficient high impact chemistry, and lacks the urgency to publish as a communication.

Reviewer #1 (Remarks to author):

Comments:

During the past years, the field of organic electrocatalysis has undergone a renaissance and has attracted considerable attention from the chemistry community for the development of sustainable bond-forming platforms. The manuscript by He and Huang reports a method for electro-oxidative C(sp³)-H bond functionalization of alkyl nitriles to furnish sulfur-containing β -enaminonitrile derivatives. I think that the manuscript describes a rather unique chemical transformation that distinguishes the work from a majority of previously developed electrocatalytic bond-forming protocols. Furthermore, the developed transformation provides potentially synthetically valuable products; however, the synthetic utility should be further highlighted. Therefore, I believe that the manuscript could be a nice addition to researchers working in the electrocatalysis community, and therefore support publication of the work in *Nature Communications* if the concerns and technical issues described below can be successfully addressed.

Concerns:

- 1) Can the developed protocol be extended to other alkyl nitriles or is it limited to acetonitrile?
- 2) The synthetic utility of the products should be further highlighted, i.e. what can, for example, compound **2aa** be transformed into?
- 3) The role of 1,2-bis(diphenylphosphino)ethane (DPPE) needs to be further investigated/clarified in order to rule out coordination to free metal ions, i.e. acting as a ligand.
- 4) Also, the authors state that "it was tentatively proposed that the corresponding oxide of DPPE might have coordinated with the intermediate through the hydrogen-bonding interactions, which renders the formation of Z-isomer more favorable". However, it is not clear how/in what way the potentially oxidized DPPE facilitates formation of the Z-isomer. Please highlight this with a figure and

if possible provide, for example, calculations that support this hypothesis.

Comments on style/writing issues:

5) I strongly suggest the authors to use colors in the figures as this would make the figures more attractive.

6) Some language corrections are necessary (please see the attached pdf - file).

Our response:

We are grateful to the reviewer for the recommendation and the suggestions.

1) Can the developed protocol be extended to other alkyl nitriles or is it limited to acetonitrile?

The reactions using other alkyl nitriles did not give the desired products. As can be seen from the following equation, if R is a carbon chain instead of a H atom, the intermediate imine **A** could not be isomerized to its enamine form, the structure **B**. As the imine **A** is not stable, the whole reaction pathway is not favored.

2) The synthetic utility of the products should be further highlighted, i.e. what can, for example, compound **2aa** be transformed into?

The synthetic utility of this methodology was further investigated and the results are included in the manuscript and supplementary information. 4H-1,4-benzothiazine scaffolds were obtained by copper-catalyzed cyclization of **2bb** and the corresponding derivatives (**3a** and **3b**) in the conversion yields of 70-87%. Notably, 4H-1,4-benzothiazine scaffolds are widely used in pharmaceutical chemistry due to their activities of antimicrobial, anticancer and so on.

Fig. Product transformations. ^aReaction conditions: (i) acetyl chloride, Et₃N, CH₂Cl₂, 0 °C to reflux, 12 h, 85%; BnBr, NaH, dry DMF, N₂, 0 °C to r.t., 4 h, 79%. (ii) CuI, K₂CO₃, *trans*-*N,N'*-dimethylcyclohexane-1,2-diamine, *N,N'*-dimethylethylenediamine, toluene, N₂, 120 °C, 15 h, conditions to be optimized. ^bConversion of substrates: **3c**, 60%; **3d**, 72%; **3e**, 62%.

3) The role of 1,2-bis(diphenylphosphino)ethane (DPPE) needs to be further investigated/clarified in order to rule out coordination to free metal ions, i.e. acting as a ligand.

Experimental results have shown that DPPE was oxidized to DPPE-oxide at the beginning of the reaction and in the presence of DPPE-oxide, the selectivity was increased. The role of DPPE-oxide has been studied by a calculation and the detail of the calculation results has been described in the following question 4. Besides, no transition metal was added to the reaction system, the possibility of the coordination of free metal ions with DPPE-oxide is very low.

4) Also, the authors state that “it was tentatively proposed that the corresponding oxide of DPPE might have coordinated with the intermediate through the hydrogen-bonding interactions, which renders the formation of Z-isomer more favorable”. However, it is not clear how/in what way the potentially oxidized DPPE facilitates formation of the Z-isomer. Please highlight this with a figure and if possible provide, for example, calculations that support this hypothesis.

We further carried out density functional theory (DFT) calculations to provide insights into the mechanism and these results have been added in the manuscript and supplementary information. DFT results indicate that imine type radical intermediates (**10** or **10'**) are more stable than the enamine type radicals (**9** or **9'**). The imine radical would interact with DPPE-oxide to form complexes **C3-C5**, among which **C3** is the most stable one with the calculated formation energies of -1.9 kcal/mol. The complex **C3** can stabilize the yielded radical and facilitate the C-S bond formation in Z configuration. It should be noted that the enamine type products (**2aa**) are more stable than imine types (**11** and **11'**). Therefore, the formed imine products would tautomerize to give the desired product **2aa**, in which the (**Z**)-**2aa** is more stable than the (**E**)-**2aa** by 1.9 kcal/mol. The predicted Z:E is around 25:1, which is in excellent agreement with our experimental observation (19:1). Considering the relative stability between (**Z**)-**2aa** and (**E**)-**2aa**, the reaction should be thermodynamic control. DFT results suggest the important role of DPPE-oxide in stabilizing the imine radical **10**, facilitating the formation of Z product and its tautomerization to final product (**Z**)-**2aa**.

Fig. DFT study of the key intermediates and the calculated structures (Å, red) for complex **C3**. The free energies are reported in kcal/mol at the M06-2X/6-311++g (d, p)/SMD(Acetonitrile) level of theory.

5) I strongly suggest the authors to use colors in the figures as this would make the figures more attract

Colors have been used in the figures.

6) Some language corrections are necessary (please see the attached pdf - file).

We have corrected them according to the reviewer's suggestions listed in the attached pdf file.

Reviewer #2 (Remarks to author):

This manuscript reports a radical initiated C(sp³)-H bond oxidative functionalization of alkyl nitriles through a KI mediated indirect anodic oxidation. High chemoselectivities and good stereoselectivities of the reactions could be achieved under the metal-free, external chemical oxidant-free conditions. A reliable mechanism is proposed after some control experiments and cyclic voltammetry experiments were conducted. The results described by the authors are informative and interesting to Nature Communications readers, particularly those working on the electroorganic chemistry. Based on the content and quality of this paper, I think this work is suitable for *Nature Communications* however some issues should be addressed before the accepting this manuscript for publication.

- 1) The reaction conditions have been optimized (constant current conditions - it would be useful to also report the potential range from the beginning to the end of these electrosyntheses) in an undivided cell.
- 2) Some information on the electrode choice is required. Can the reaction also be performed with a carbon working electrode? More cheap electrodes should be examined.
- 3) In the proposed mechanism, radical intermediate **9** could substitute with the disulfide **11** to produce an imine **12**, the authors should give the corresponding evidence to prove this process. A reaction of disulfide **11** with acetonitrile should be conducted.
- 4) In order to make the mechanism more convincing, authors should carry out more cyclic voltammetry (CV) experiments. For example, the CV experiments of a mixture of **1aa**, acetonitrile, and KI should be done.
- 5) Table 1, entry 2 and entry 3, the author's description does not conform to the table.

Our response:

We are grateful to the reviewer for the recommendation and the suggestions.

- 1) The reaction conditions have been optimized (constant current conditions - it would be useful to also report the potential range from the beginning to the end of these electrosyntheses) in an undivided cell.

The potential range from the beginning to the end is about 2.8 ~ 5.5 V.

- 2) Some information on the electrode choice is required. Can the reaction also be performed with a carbon working electrode? More cheap electrodes should be examined.

The results of other electrode choice have been added in supplementary information.

entry	Variation from the standard conditions	yield(%) ^[b]	Z/E ^[b]
-------	--	-------------------------	--------------------

1	none	96	19:1
2	RVC as an anode ^[c]	95	14:1
3	Carbon plate as an anode ^[c]	95	12:1
4	Carbon rod as an anode ^[c]	94	14:1
5	Carbon cloth as an anode ^[c]	91	12:1
6	Carbon cloth as a cathode ^[c]	35	11:1

^a Standard conditions: **1aa** (0.5 mmol), citric acid (10 mol %), DPPE (20 mol %), KI (50 mol %), MeCN (5 mL), with 0.1 M *n*-Bu₄NClO₄ as electrolyte. A Pt minigrad electrode (52 mesh, 1 × 1.5 cm²) as an anode and a Pt wire (diameter = 0.5 mm, height = 2.0 cm) as a cathode, an undivided cell, constant current = 10 mA, 4 h, room temperature, 3.0 F/mol. ^b Yields and *Z/E* ratios were determined by ¹⁹F NMR analysis of the crude reaction mixture using fluorobenzene as the internal standard. ^c RVC anode (100 PPI, 1.0 × 1.5 × 1.0 cm³), Carbon rod anode (diameter: 0.5 cm, height: 1.78 cm), Carbon cloth anode (1.0 × 1.5 cm²), Carbon cloth cathode (0.3 × 1.5 cm²).

3) In the proposed mechanism, radical intermediate **9** could substitute with the disulfide **11** to produce an imine **12**, the authors should give the corresponding evidence to prove this process. A reaction of disulfide **11** with acetonitrile should be conducted.

This process had been proven by the reactions with phenyl disulfide **1g**, phenyl diselenide **1p** and methyl diselenide **1q** as substrates (see Fig. 2, **2g**, **2p** and **2q**), and in which, yields of 72%, 92%, and 91% were obtained, respectively.

4) In order to make the mechanism more convincing, authors should carry out more cyclic voltammetry (CV) experiments. For example, the CV experiments of a mixture of **1aa**, acetonitrile, and KI should be done.

The CV experiments of a mixture of **1aa**, acetonitrile, and KI have been carried out and these results are included in supplementary information. Cyclic voltammetry studies on the mixture of **1aa**, acetonitrile, and KI (Fig. B, curve c) also exhibited obvious catalytic currents compared with only KI in the acetonitrile solution (Fig. B, curve b). In addition, the oxidation peak of **1aa** was observed at 1.85 V vs. SCE (Fig. B, curve d).

Fig. B. Cyclic voltammograms of 0.1 M *n*-Bu₄NClO₄ solution in MeCN at room temperature. (a) None; (b) KI (50 mmol/L); (c) KI (50 mmol/L) + **1aa** (2 mmol/L), potential range from -0.5 to 1.2 V; (d)

KI (50 mmol/L) + **1aa** (2 mmol/L), potential range from -0.5 to 2.0 V. The voltammogram was obtained with Pt wire as an auxiliary electrode and a saturated calomel electrode (SCE) as a reference electrode. The scan rate was 0.1 V/s on a platinum disk electrode (d = 2 mm).

5) Table 1, entry 2 and entry 3, the author's description does not conform to the table.

We have corrected them.

Reviewer #3 (Remarks to author):

In the present manuscript, Huang and He develop a method for electro-oxidative C(sp³)-H bond functionalization of acetonitrile to synthesize sulfur-containing beta-enaminonitriles. While the work is interesting, it is not breaking new ground at a level for publication in *Nature Communications*. Although several functional groups were contained in the products, such products are lack of universality. As it stands the work is incomplete in relation to limited utility of the processing of aliphatic thiols. Except for acetonitrile, no other alkyl nitriles were tested in the manuscript. As a research on electrochemical synthesis, the authors do not explain why the reaction cannot occur when LiClO₄ and NaClO₄ were used as the electrolyte. This is not a common phenomenon. Most importantly, the manuscript contains insufficient high impact chemistry, and lacks the urgency to publish as a communication.

Our response:

We are grateful to the reviewer for the comments.

1) As it stands the work is incomplete in relation to limited utility of the processing of aliphatic thiols.

This methodology has demonstrated its good substrate scope with up to 27 examples. Most of the general functional groups (including a hydroxy group) and heteroaromatics are tolerated. For the aliphatic thiols, the yield is up to 53% for 2-phenylethanethiol, although the yields are low for 1-hexanethiol and cyclohexanethiol. It is noteworthy that excellent yield (up to 91%) has been observed for the aliphatic diselenide.

For the 1-hexanethiol and cyclohexanethiol, it is tentatively suggested that the reactivity of the formed aliphatic disulfides under the electrochemical conditions toward the intermediate **10** is lower than that of aromatic disulfides (Please see Ref. 65 in the manuscript).

2) Except for acetonitrile, no other alkyl nitriles were tested in the manuscript.

Other alkyl nitriles have been tested. Kindly please see the response to reviewer 1, question No. 1.

3) As a research on electrochemical synthesis, the authors do not explain why the reaction cannot occur when LiClO₄ and NaClO₄ were used as the electrolyte. This is not a common phenomenon.

For the majority of the electrochemical synthesis, the choice of an electrolyte is simple and usually bases on the solubility of the electrolyte in the solution. However, the choice of the electrolyte which shows a direct impact on the course of a reaction has been reported in some works, for examples, *Org. Lett.* **2010**, *12*, 2590–2593, *Org. Lett.* **2011**, *13*, 1678–1681, and *Chem. Commun.*, **2013**, *49*, 8982–8984. These researches showed the difference of the effects on the reaction activities between using a quaternary ammonium salt and a Li /Na salt as the electrolyte. Recently, Professor Moeller presented a discussion about this phenomenon in a review (Please see Page 4821 of Ref. 31 in the manuscript). The nature of the electrochemical “double layer” (or outer Helmholtz layer for a physical chemist) at the anode surface formed from a quaternary ammonium salt is different from that of a Li /Na salt. At the current stage, we proposed that the reactivity of our reaction system was affected by the double layers formed on the anode from different electrolytes.

REVIEWERS' COMMENTS:

Reviewer #1 (Remarks to the Author):

In the revised manuscript the authors show the synthetic utility of the obtained sulfur-containing β -enaminonitrile products. Furthermore, the role of DPPE has now been investigated through DFT calculations, thus supporting the proposed mechanism and providing a plausible explanation of the observed selectivity. With the current additions I believe that the revised manuscript will be a nice addition to the electrochemistry community, and therefore support publication of the work in Nature Communications.

Reviewer #2 (Remarks to the Author):

This reviewer found that the authors have totally revised their submission, in the light of the reviewers' comments. Additional control experiments, DFT calculations were well done, which would convince the readership of Nature Communications.

This reviewer recommends this for publication in Nature Communications as it stands.

Reviewer #1 (Remarks to author):

Comments:

In the revised manuscript the authors show the synthetic utility of the obtained sulfur-containing β enaminonitrile products. Furthermore, the role of DPPE has now been investigated through DFT calculations, thus supporting the proposed mechanism and providing a plausible explanation of the observed selectivity. With the current additions I believe that the revised manuscript will be a nice addition to the electrochemistry community, and therefore support publication of the work in Nature Communications.

Our response:

We are grateful to the reviewer for the nice comments of our work and for the recommendation.

Reviewer #2 (Remarks to author):

This reviewer found that the authors have totally revised their submission, in the light of the reviewers' comments. Additional control experiments, DFT calculations were well done, which would convince the readership of Nature Communications.

This reviewer recommends this for publication in Nature Communications as it stands.

Our response:

We are grateful to the reviewer for the nice comments of our work and for the recommendation.